# Safety Evaluation and Whole Genome Sequencing of *Aspergillus japonicas* PJ01 Reveal Its Potential to Degrade Citrus Segments in Juice Processing

**DOI:** 10.3390/foods10081736

**Published:** 2021-07-27

**Authors:** Yujiao Qian, Zhipeng Gao, Jieyi Wang, Chen Wang, Gaoyang Li, Fuhua Fu, Jiajing Guo, Yang Shan

**Affiliations:** 1Longping Branch, Graduate School of Hunan University, Changsha 410125, China; yujiaoqian2021@163.com (Y.Q.); wangjieyi1996@163.com (J.W.); ch13750@163.com (C.W.); lgy7102@163.com (G.L.); fhfu686@163.com (F.F.); 2International Joint Lab on Fruits & Vegetables Processing, Quality and Safety, Hunan Key Lab of Fruits & Vegetables Storage, Processing, Quality and Safety, Hunan Agriculture Product Processing Institute, Hunan Academy of Agricultural Sciences, Changsha 410125, China; 3College of Animal Science and Technology, Hunan Agricultural University, Changsha 410128, China; gaozhipeng627@163.com

**Keywords:** *Aspergillus japonicas* PJ01, segments degradation, gizzard juice process, safety evaluation, genome sequencing

## Abstract

*Aspergillus japonicas* PJ01 (*A. japonicas* PJ01) is a strain isolated from the rotten branches. In previ-ous studies, it was shown that it can produce complex enzymes to degrade polysaccharide com-ponents. In this study, we evaluated the safety of its crude enzyme solution. Acute oral toxicity, subchronic toxicity, micronucleus and sperm malformation tests all validated the high biologi-cal safety for the crude enzymes. Secondly, we carried out the citrus segment degradation ex-periment of crude enzyme solution. Compared with the control group, the crude enzyme solu-tion of *A. japonicas* PJ01 can completely degrade the segments in 50 min, which provides the basis for enzymatic peeling during juice processing. The whole genome sequencing showed that the genome of *A. japonicus* PJ01 has a GC content of 51.37% with a size of 36204647 bp, and encoded 10070 genes. GO, COG, KEGG and CAZy databases were used in gene annotation analyses. Pathway enrichment showed many genes related to carbohydrate metabolism, rich in genes re-lated to pectinase, xylanase and carboxylcellulase. Therefore, the complex enzyme produced by *A. japonicus* PJ01 can be used in gizzard juice processing to achieve efficient enzymatic decapsu-lation.

## 1. Introduction

China is a main producing area of citrus, and the fruit juice processing industry is huge. With the change of public health awareness, fruit juice has changed from traditional juice drinks to gizzard juice with fruit grains, the most famous of which is “Minute Maid”, which has complete citrus cysts. The cysts used in the production of Minute Maid orange beverage are mainly produced by the acid–base method. However, acid–base treatment easily produces heavy metal pollution. It will also produce a large number of broken fragments, which will have adverse effects on the senses [1]. In recent years, the application of enzymatic decapsulation has attracted great attention and has achieved good results in the production of citrus cans. The principle of peeling by enzyme preparation is to digest the substances existing in the cell wall [2]. The key problem in citrus juice production is how to remove the segments with high quality and low cost.

The chemical composition of the cortex mainly includes pectin, cellulose, hemicellulose etc. Pectinase treatment can reduce the integrity of segments [3]. The traditional enzyme preparation usually uses glucose, pectin, xylan and other carbon sources as substrates, which costs too much. In recent years, increasingly more studies have examined ways to use agricultural wastes such as bran and citrus dregs, which can reduce costs and alleviate environmental pollution [4]. Plant biomass is degraded by fungi that can produce carbohydrate-active enzymes to degrade plant polysaccharides [5]. *Aspergillus* spp. has been proven to produce cellulase, hemicellulose and pectinase, and can be used to clarify fruit juice [6]. Xylanase can also improve the clarity of juice [7,8]. The mixed enzyme preparation produced by *Aspergillus niger* can achieve a better peeling effect on *citrus paradisi* peel by vacuum perfusion [9]. Generally, *Penicillium* and *Aspergillus* can generate enzymes for the degradation of lignocellulose [10]. Research shows that new strains of *black Aspergilli* can produce multiple hydrolytic enzymes, including cellulase, pectinase and tannase [11]. The effect of a single enzyme is often not very good, the treatment time is long and the degradation effect of a capsule is not ideal, so screening strains with a high yield of compound enzyme has a higher utilization value. Most commercial enzymes need to be compounded, and the cost of separation and purification of commercial enzymes is high. However, the fungal complex enzymes produced by fermentation of orange peel residue and other wastes contained high-quality complex enzyme system for degrading cellulose, hemicellulose and pectin, which has great application potential.

A part of *Aspergillus* has toxicity, similar to *Aspergillus flavus*, which has strong carcinogenic toxicity. The safety of some strains with similar enzyme production ability has not been confirmed, such as *Aspergillus niger* and *Penicillium oxalicum* [9,12]. Food enzyme preparation is required to have no potential safety hazard, whether *A. japonicus* PJ01 and its enzyme can be applied to food production must have a safety evaluation to be convincing [13]. In order to better understand the biological information of fungi and bacteria, whole genomes were sequenced. The genome sequencing of *Fibrobacter succinogenes* revealed how it degraded cellulose and adapted to rumen environment [14]. The complete genome sequence can be used to understand the enzyme system of cellulose degradation [15]. Genome sequencing is convenient for genetic engineering technology to develop more efficient and active microorganisms.

The purpose of this study is to understand the enzyme production ability of *A. japonicas* PJ01 used in juice processing. In previous studies, we screened filamentous fungi with a high yield of complex enzymes, which resulted in the selection of *A. japonicas* PJ01 obtained from rotten branches. In order to predict the enzyme-producing gene fragment, the whole genome was sequenced. The degradation effect of crude enzyme produced by fungi on citrus segments was observed. The safety of crude enzyme was evaluated by animal experiments. It provides a theoretical basis for the realization of low-cost, high-efficiency and safe enzymatic peeling method in fruit and vegetable juice process.

## 2. Materials and Methods

### 2.1. Fungal Strain

The fungal strain used in this study was *A. japonicas* PJ01 (CCTCC NO: M2013323), isolated from the rotten branches in Yuelu Mountain soil. It was kept in the China Center for Type Culture Collection (CCTCC).

### 2.2. Preparation of Crude Enzyme Solution

Production and extraction methods of the enzyme refer to previous experimental studies [16]. *Aspergillus japonicus* PJ01 was cultured at 35 °C for three days, and then washed with normal saline containing 0.1% Tween 80 to prepare spore suspension (1 × 10^6^/mL). Using 75 mL modified Czapek’s medium (g/L) (NaNO_3_ 5, K_2_HPO_4_ 1, KCL 0.5, MgSO_4_·7H_2_O 0.5 and FeSO_4_·7H_2_O 0.01), 1% orange peel powder and 0.35 g PEG4000, 1 mL spore suspension was inoculated in it after high pressure sterilization. It was then cultured at 35 °C for 72 h. The supernatant after filtration and centrifugation was used as crude enzyme solution. The enzyme activity was determined under submerged fermentation in the previous study. The maximal pectinase, carboxyl cellulase (CMCase) and xylanase activities reached 2610, 85 and 335 U/gds (units/gram dry substrate) after 72 h of cultivation [17].

### 2.3. Degradation of Citrus Segments

The degradation effect of crude enzyme solution on citrus segments was observed. A 50 mL centrifuge tube was used as the reaction vessel. After the orange slices were peeled off by hand, the orange petals with relatively uniform size and quality were used to remove the collaterals. The orange slices were covered with plastic film for standby. The ratio of crude enzyme solution to orange slice was 1:5 (*w*/*w*). The degradation reaction temperature is 50 °C in the water bath. The centrifuge tube was removed every ten minutes and gently shaken to ensure full reaction. The degradation was observed at 10, 20, 30, 40 and 50 min. The control group (CK) did not do any treatment. Macrodegradation pictures of orange slices were taken at different times. The segment fragments were sectioned, stained with toluidine blue and observed under an upright optical microscope (NIKON ECLIPSE E100).

### 2.4. Safety Evaluation of Crude Enzyme Solution

All animals used in this study were provided by Hunan Sileike Jingda Co. (Changsha, China). The study was reviewed and approved by the Institutional Animal Care and Use Committee (IACUC) of Hunan Sileike Jingda Co. (Changsha, China), certificate IACUC-SJA18072. The animals were housed in a controlled room with a 12 h/d lighting cycle during the experimentation. The following tests in mice were evaluated to gain insight concerning the safety of the crude enzyme solution: acute oral toxicity, bone marrow micronucleus, sperm deformity and subchronic toxicity feeding. Four independent experiments were performed.

#### 2.4.1. Acute Oral Toxicity Test

Forty SPF ICR mice (20 + 2 g), half male and half female, were fed at maximum tolerance for 14 days. Mice were randomly divided into two groups, the control group and the enzyme liquid group (*n* = 20). Mice were given enzyme solution at 40 mL/kg once a day by oral gavage according to Horne’s method. The control group was given same volume of pure water. After administration, the mice were kept under observation for 0–4 h, lasting for 14 days. The daily observations of the mice included appearance, behavioral activities, secretions, excretions, diet and death (time of death, predeath reaction). The body weight and deaths were recorded on the 4th, 7th, 10th and 14th day after the treatment [18].

#### 2.4.2. Micronucleus Test of Bone Marrow Cells

Using the method of 30 h administration of test substance, 50 ICR mice, half male and half female, weighing 18–25 g, were randomly divided into five groups according to their body weight (control group (C), low (L), medium (M) and high (H) dose enzyme solution groups, and positive control group (P), *n* = 10). The positive control group was intraperitoneally injected with cyclophosphamide at a dose of 40 mg/kg, and the volume of each test substance was 20 mL/kg. The high-dose group was given the original enzyme solution, the middle-dose group was one-half of the high-dose, the low-dose group was one-quarter of the high-dose, and the control group was given pure water by gavage. The same administration was repeated again 24 h later. Six hours after the last administration, the mice were killed by cervical dislocation and the femoral bone marrow smear was taken. Under the light microscope, 1000 polychromatic erythrocytes (PCES) were counted in each animal. The micronucleus rate was calculated by the percentage of PCE-containing micronuclei; 200 polychromatic erythrocytes were counted and the ratio of polychromatic erythrocytes to mature erythrocytes (PCE /NCE) was calculated [19,20].

#### 2.4.3. Sperm Abnormality Test

Six-week-old male ICR mice were randomly divided into five groups according to their body weight, including blank control group, low-, medium- and high-dose enzyme solution groups and positive control group, with 5 mice in each group. The positive control group was intraperitoneally injected with cyclophosphamide at a dose of 40 mg/kg, and the treatment dose of other groups were the same as those in Section 2.4.2. On the 35th day after the first administration of the test substance, the mice were killed by cervical dislocation, and 1000 sperm with complete structure were counted for each animal. The incidence of abnormal sperm was calculated [21].

#### 2.4.4. Subchronic Toxicity Test

Four-week-old SD rats, half male and half female, were selected and randomly divided into control and enzyme solution groups according to body weight. The test group was given the crude enzyme solution by gavage, and the control group was given the same volume of pure water by gavage. The volume was 10 mL/kg, administered once a day. After 28 days, blood samples were collected for hematological and biochemical tests. The weights of liver, kidney, spleen and testis were weighed, and the ratio of organ to body was calculated [22].

### 2.5. Whole Genome Sequencing, Assembly and Annotation

*A. japonicas* PJ01 has the ability to produce pectinase, CMCase and xylanase. The study obtained the corresponding dominant phenotype specific genes through genome sequencing and genome annotation analysis. Genomic DNA was extracted using the Omega Fungal DNA Kit D3390-02 according to the manufacturer’s instructions. Purified genomic DNA was quantified by a TBS-380 fluorometer (Turner BioSystems Inc., Sunnyvale, CA, USA). High quality DNA (OD260/280 = 1.8–2.0, >15 ug) was used to do further research.

The genomic DNA was built into a 10 kb template library and sequenced using a paired-end (2 × 150 bp) sequencing methodology with PacBio Sequel Single Molecule Real Time (SMRT) and Illumina sequencing platforms. DNA samples were cut into 400–500 bp fragments by using the Covaris M220 Focused Acoustic Shearer. The Illumina sequencing library was prepared from the cut fragments with the NEXTflexTM Rapid DNA-Seq Kit [23,24]. Illumina sequencing data were assembled with Velvet. Cegma 2.5 and Busco 3.0 were used for quality assessment of genome assembly. The sequence data, defined as raw data or raw reads, were the conversion of the original image data through base calling, and were saved as FASTQ files, which are raw data provided to users, including read order and quality information. The method of quality information statistics was used for quality trimming, removing low-quality data, and forming clean data. After subread filtering of raw data from the PacBio RS II and Illumina PE150 sequencer, the reads were then assembled into contigs using CANU. [25]. Possible misassemblies were corrected with the gap resolution software or sequencing cloned bridging PCR fragments with subcloning or transposon bombing. Bioinformatics analysis was performed using data generated by the Pacbio and Illumina platforms. I-Sanger Cloud Platform (www.i-sanger.com, accessed on 17 May 2021) from Shanghai Majorbio was used to conduct all the analyses. The reads were used after quality control to assemble from scratch, and the assembly quality of samples counted by using Maker 2, which performed coding sequence (CDS) prediction. The Barrnap 0.4.2 and Trna scan-SE v1.3.1 software were used to predict the rRNA and tRNA contained in the genome. The predicted CDSs were annotated from Kyoto Encyclopedia of Genes and Genomes (KEGG), Clusters of Orthologous Groups (COG), Gene Ontology (GO) and Carbohydrate-Active Enzymes (CAZy) database by using sequence alignment tools such as BLAST_+_ 2.3.0, Diamond 0.8.35 and Hmmer 3.1b2.

### 2.6. Statistical Analysis

Statistical analysis was performed by SPSS software 26.0 and Excel. The results were expressed as mean ± SD. The measurement data were tested for normality and homogeneity of variance. If the normality (*p* > 0.05) was satisfied, one-way ANOVA was used for statistical analysis, and LSD + Dunnet or Tamhane’s T2 was selected for comparative analysis according to the homogeneity of variance. If the normality was not satisfied (*p* ≤ 0.05), the Mann–Whitney U test was used for pairwise comparison analysis, where *p* ≤ 0.05 means statistically significant and *p* ≤ 0.01 means highly statistically significant. The online platform of Major bioCloud (http://www.majorbio.com/) (accessed on 17 May 2021) was used to analyze the data of whole genome sequencing.

## 3. Results

### 3.1. Effect of Crude Enzyme Solution on Degradation of Citrus Segments

As shown in Figure 1, in order to observe the degradation effect of crude enzyme solution on citrus segments, the degradation study was carried out. From the appearance of orange petals, it can be seen that with the extension of treatment time, the outer layer of the capsule gradually becomes thinner and transparent. From the beginning, the complete thickness gradually degraded to the end, when it was completely degraded, revealing the complete juice cells. The microscopic and macroscopic images showed the same trend. In the control group, the citrus segment had a certain thickness, compact arrangement, clear cell edge, complete and solid cell wall, and uniform intercellular material. With the extension of treatment time, the capsule kept intact cell morphology from 10 min, and then began to deform slightly. After 20 min, part of cell wall and cell membrane disappeared, and the intercellular material began to be uneven, resulting in cell fusion. After 30 min, the cells began to break up, the fibers appeared as holes, and the intercellular layer peeled off. After 40 min, the intercellular material completely disappeared, and the cells completely or partially disintegrated. At the end (50 min) only some fragments were left, and there was no complete cell structure.

### 3.2. Acute Oral Toxicity Test in Mice

The animals in control group and enzyme solution group were observed continuously for 14 days after gavage, and no abnormal reaction or death was found. At the end of the experiment, gross anatomy was conducted on the mice, and no obvious abnormal conditions were found on the surface and section of main organs. Compared with the control group, there was no significant difference in body weight between the enzyme solution group and the blank control group before administration (*p* > 0.05). The body weight of the enzyme group was not significantly different from that of the control group on the 4th, 7th, 10th and 14th day after the administration (*p* > 0.05). The maximum tolerated dose of *A. japonicas* PJ01 crude enzyme solution in ICR mice was more than 40 mL/kg (Figure 2).

### 3.3. Micronucleus Test of Mice Bone Marrow Cells

The micronucleus rate of mice bone marrow cells in each dose group of enzyme solution was not significantly different from that of the control group (*p* > 0.05), but there was significant difference between the positive control group and control group (*p* < 0.01). PCE/NCE ratio of each dose group was not less than 20% of the control group, indicating that the enzyme solution had no obvious toxicity to mice bone marrow cells (Table 1).

### 3.4. Sperm Deformity Test

There was no significant difference in sperm deformity rate between the enzyme solution groups and the control group, but there was significant difference between the positive control group and the control group (*p* < 0.01) (Table 2).

### 3.5. Subchronic Toxicity Test

#### 3.5.1. The Influence of Enzyme Solution on Body Weight and Food Utilization Rate in Rats

During the 28-day feeding period, the rats were weighed twice a week and weighed at the end of the 28-days feeding. There were no significant differences in body weight and weekly food utilization rate between the animals and the control group (*p* > 0.05) (Figure 3 and Figure 4).

#### 3.5.2. Effect of Enzyme Solution on Hematological Parameter Analysis of Rats

The hematological analysis confirmed there was no significant difference in indicators of the PLT, HGB, HCT, WBC NEU, LYM, MON, BAS and RBC between the enzyme solution and control groups (*p* > 0.05) (Table 3). For males, the EOS of the enzyme group was significantly different from that of the control group (*p* < 0.05), but it was within the normal range and had no physical significance.

#### 3.5.3. Effect of Enzyme Solution on Blood Biochemistry of Rats

There was no significant difference in TP, ALB, BUN, GLU, ALT, AST, CHO, TG or Cr between the enzyme solution group and the control group (*p* > 0.05) (Table 4).

#### 3.5.4. Effects of Enzyme Solution on Organ Weight and Organ Body Ratio in Rats

There was no significant difference in the weight of liver, kidney, spleen and testis and the ratio of viscera to body between the enzyme solution group and the control group in female and male rats (*p* > 0.05) (Figure 5).

### 3.6. Genome Features of A. Japonicas PJ01

The complete genome of *A. japonicas* PJ01 consists of one 36204647 bp chromosome and 10070 protein cording sequences (CDS), 351 tRNA genes, 93 rRNA genes and an average G + C content of 51.37% (Table 5). The outermost layer of the circle is the scaffold arrangement, and the second vertical line represents all secretory glycoside hydrolases, including glucoside hydrolase, pectinase, carboxycellulase and xylanase. The third circle is the GC content. GC skew value was the innermost value, which was more likely to transcribe CDS (Figure 6).

### 3.7. COG Analysis

From COG analysis, there are 23 protein functions in the database. We can see that the most abundant COGs known function in *A. japonicas* PJ01 are (in decreasing order): carbohydrate transport and metabolism (G), posttranslational modification, protein turnover, chaperones (O), amino acid transport and metabolism (E), energy production and conversion (C). Among them, G is the largest (Figure 7).

The most abundant COGs are COG0477(major facilitator Superfamily), COG1472(hydrolase family 3), COG3325(chitinase), COG0702(epimerase dehydratase), COG0366(alpha amylase, catalytic), COG2273(hydrolase family 16) and COG2730(Glycoside hydrolase Family 5. Among them, gene04885 and gene06630 are related to endo-14-beta-xylanase in carbohydrate transport and metabolism.

### 3.8. GO Database Annotation

GO classification statistics returned 5955 unique genes, which were annotated in major categories of molecular function, cellular component and biological process (Figure 8). The number of GO items and genes in the three categories follow: molecular function (Gene number: 4465), biological process (Gene number: 4346) and cellular component (Gene number: 3723). Genes in the biological process category were divided into 23 subfunctions, among which metabolic process (GO:0008152, gene number 3239), cellular process (GO:0009987, 2547) and single-organism process (GO:0044699, 2085) were the most; the cellular component category contained 14 subfunctions of genes, most of which were related to functions of the cell (GO:0005623, 2250), cell part (GO:0044464, 2237) and membrane (GO:0016020, 1960); the molecular function category had 13 subfunctions of genes, most of which involved catalytic activity (GO:0003824, 3194) and binding (GO:0005488, 2268). According to the GO annotation of *A. japonicas* PJ01, genes in the metabolic process (GO:0008152), catalytic activity (GO:0003824) and cellular process (GO:0009987) subfunctions were the three most abundant.

### 3.9. KEGG-Pathway Annotation

As shown in Figure 9, KEGG classification statistics returned 3303 genes. There were six classifications of KEGG pathways: metabolic, human diseases, organismal systems, genetic information processing, environmental information processing and cellular processes. Three hundred thirty-nine genes were assigned to the carbohydrate metabolism, followed by the amino acid metabolism with 284 genes, and global and overview maps with 262 genes. Fifty-nine genes were related to Amino sugar and nucleotide sugar metabolism (ko00520). Fifty-two genes were related to starch and sucrose metabolism (ko00500), which is related to the degradation and metabolism of Acellulose 1,4-β-cellobiosidase. There are also four genes (gene04885, gene06233, gene08287, gene06630) associated with endo-1,4-β-xylanase, and four genes (gene01710, gene07268, gene10354, gene09437) associated with pectin lyase.

### 3.10. Carbohydrate Active Enzymes (CAZyme) Database Annotated of Carbohydrate Active Enzyme Gene

Compared with the CAZyme database, 601 genes were annotated. The genes of glycoside hydrolases (GH) accounted for 45.42% (273), which was the most. The genes of glycosyl transferases (GT) (99), carbohydrate esterases (CE) (103) and auxiliary activities (AA) (95) accounted for 16.47%, 17.14% and 15.81%, respectively. The complete genome of *A. japonicas* PJ01 also included 12 polysaccharide lyases (PL) and 19 carbohydrate-binding modules (CBM) (Figure 10). GH contains 12 genes related to polygalacturonase, 8 genes related to xylanase and 10 genes related to carboxymethyl cellulose.

## 4. Discussion

In this study, we comprehensively studied the genomic information of *A. japonicas* PJ01. We induced its enzyme production, evaluated the safety of its crude enzyme solution, and observed its effect in the degradation of citrus segments. The whole genome sequencing predicted the gene fragment of enzyme production.

Some *Aspergillus* can produce certain toxins, which are harmful to health [26,27]. In vivo safety evaluation can evaluate its safety in food production [28]. The acute oral toxicity test, bone marrow micronucleus test, sperm deformity test and subchronic toxicity feeding test were used to prove that the crude enzyme produced by *A. japonicas* PJ01 is safe and nontoxic.

The degradation of capsule has always been a difficult problem in juice processing and canned fruits and vegetables. Enzymatic degradation of capsule is one of the most efficient and safe methods. Studies have shown that different commercial enzyme preparations contain different decomposing enzymes, such as gelatinase, polygalacturonase and carboxymethyl cellulase, which have different effects on the degradation of segments [29]. However, there is little research on *Aspergillus*, which produces a variety of enzymes to degrade segments. However, it is not clear whether *A. japonicus* PJ01 can degrade citrus segments effectively. The application prospect of enzymatic degradation is great. It is also possible to reuse the enzyme solution in enzymatic peeling, which has more economic benefits [30].

Whole genome sequencing is used to analyze the whole genome biological information and reveal the potential function of microorganisms [31]. Genome information reveals that *C. cellulovorans* has pectate lyases, exopoly-galacturonate lyases, a pectin methylesterase and pectin esterases protein families related to degrading plant tissue [32]. It is a source of high efficiency and environmental protection to isolate and screen microorganisms that can produce industrial enzymes from the natural environment, natural plants and fruit and vegetable processing wastes. Choure found that microorganisms in alkaline hot springs have cellulase, pectinase and other enzyme activities [33]. Whole genome sequencing can provide a more comprehensive understanding of microbial diversity and function [34,35]. Endo-1,4-beta-xylanase is the main degradation enzyme of xylanase [36]. The whole genome sequencing showed that *A. japonicas* PJ01 has four genes related to endo-1,4-beta-xylanase, and its activity is stronger than that of *A. saponilacus*, which has one gene related to endo-1,4-beta-xylanase [37]. CAZy annotation showed that there are more genes in *A. japonicas* PJ01 related to carbohydrate activity than *Dickeya sp. WS52*, which can also produce pectinase, cellulase and xylanase [38]. Xylanase can be used as an auxiliary enzyme to improve the hydrolysis efficiency of cellulase [39]. *Aspergillus spp.* has high xylanase activity, such as *Aspergillus sulphureus JCM01963*, *Aspergillus fumigatus* and so on [40,41]. In KEGG annotation, many genes are assigned to carbohydrate metabolism. Genes related to the starch and sucrose metabolism (ko00500) are ranked high, which is related to pectinesterase. Pectinesterase is an enzyme that decomposes pectin polysaccharides of plant cell walls. COG analysis showed that *A. japonicas* PJ01 has many genes involved in carbohydrate transport and metabolism. COG0477 is involved in catalyzing carbohydrate transport and COG0366 is involved in encoding α -amylase [42]. Studies have shown that *A. japonicas* PJ01 can produce pectinase, xylanase and carboxylcellulase. In order to achieve efficient enzyme production and better application in fruit juice processing, the gene fragment of *A. japonicas* PJ01 can be cut and inserted into the engineered bacteria in follow-up research to realize the industrialization of efficient enzyme production.

## 5. Conclusions

The results showed that the crude enzyme produced by *A. japonicus* PJ01 contained xylanase, pectinase and carboxyl cellulase, which could effectively degrade citrus segments. It is expected to be applied in the production of sachet juice or clarification of fruit juice. The safety evaluation showed that the crude enzyme had no toxicity and could be safely used in food production and processing. The whole genome sequencing revealed the genes that produce a variety of enzymes. Through genome annotation, we found that the genes encoding xylanase in *A. japonicas* PJ01 were gene03634, gene04885, gene06233, gene08894, gene07630, gene06630, gene08287 and gene08648. The genes encoding pectinase were gene00127, gene00595, gene01769, gene03140, gene04201, gene05402, gene05448, gene06201, gene06286, gene06387, gene06464 and gene09498. Lastly, the genes encoding carboxyl cellulase were gene00744, gene01039, gene01074, gene01409, gene03135, gene03847, gene04844, gene07267, gene09062 and gene09653.

## Figures and Tables

**Figure 1 foods-10-01736-f001:**
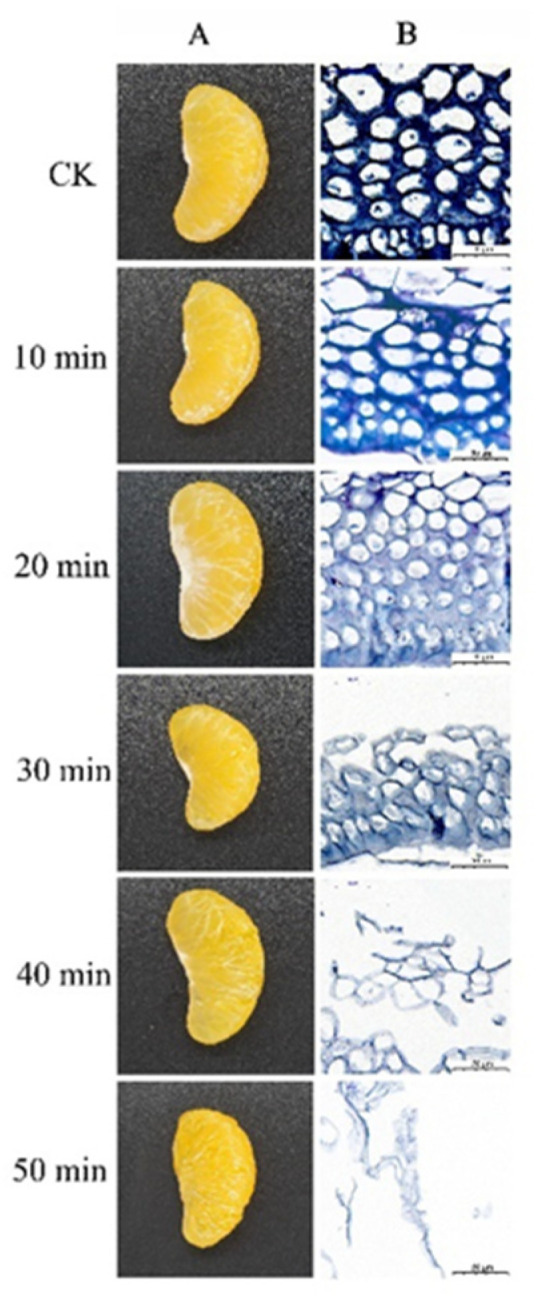
Degradation effect of the crude enzyme solution on citrus segments at subsequent times: (**A**) macroscopic observation and (**B**) microscopic observation by light microscopy. CK: control group.

**Figure 2 foods-10-01736-f002:**
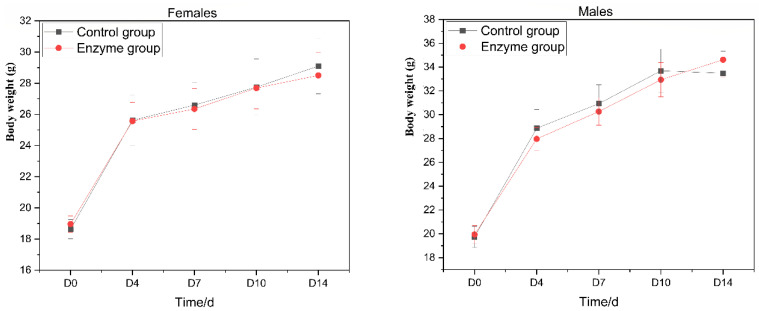
Effect of enzyme solution on body weight of mice.

**Figure 3 foods-10-01736-f003:**
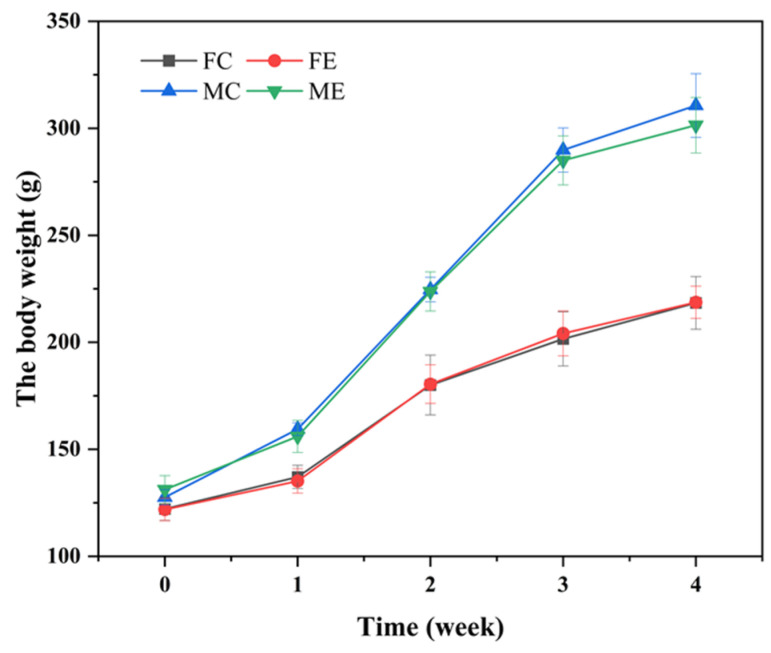
Effect of enzyme solution on body weight of rats. Groups were divided into Female control group (FC), Female enzyme group (FE), Male control group (MC), Male enzyme group (ME).

**Figure 4 foods-10-01736-f004:**
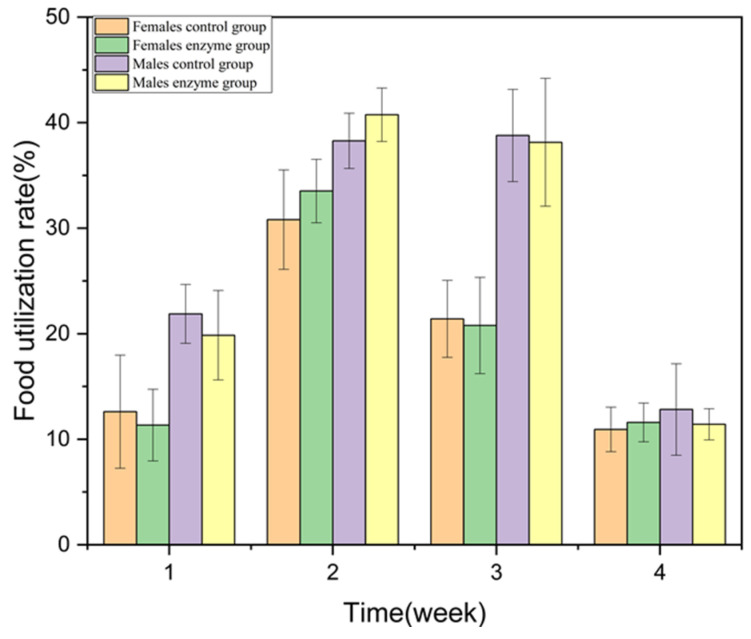
Effect of enzyme solution on food utilization rate of rats.

**Figure 5 foods-10-01736-f005:**
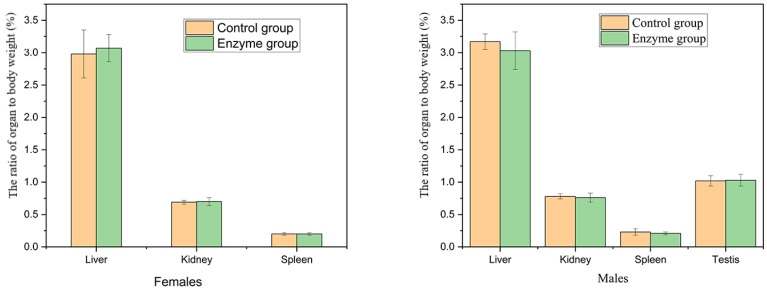
Effects of enzyme solution on organ to body weight ratio in rats (*n* = 5).

**Figure 6 foods-10-01736-f006:**
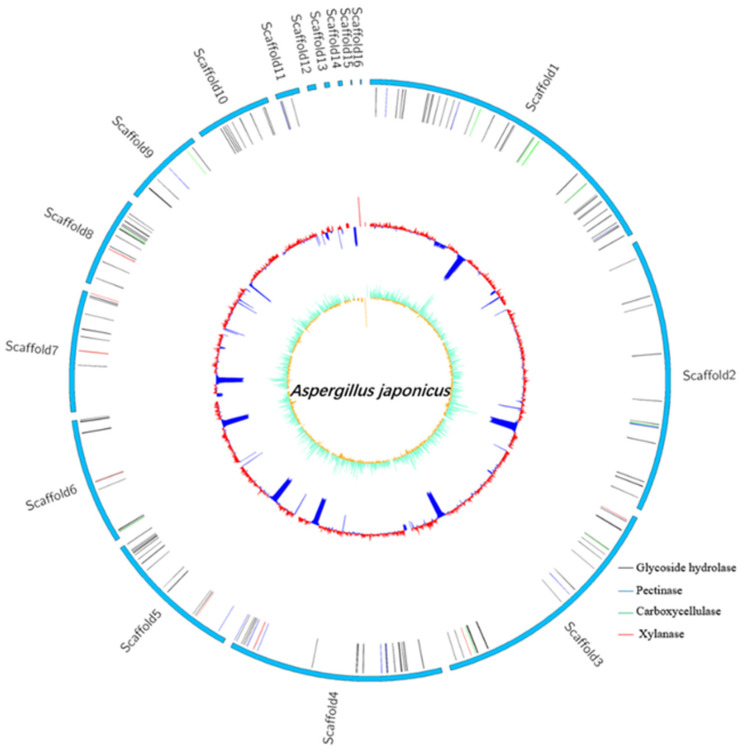
Genome map of *A. japonicas* PJ01.

**Figure 7 foods-10-01736-f007:**
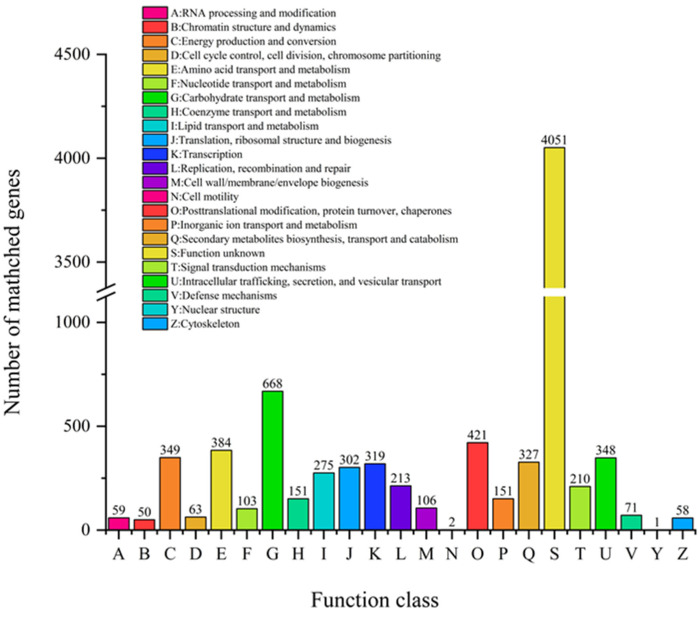
The COG function classification of *A. japonicas* PJ01 genome.

**Figure 8 foods-10-01736-f008:**
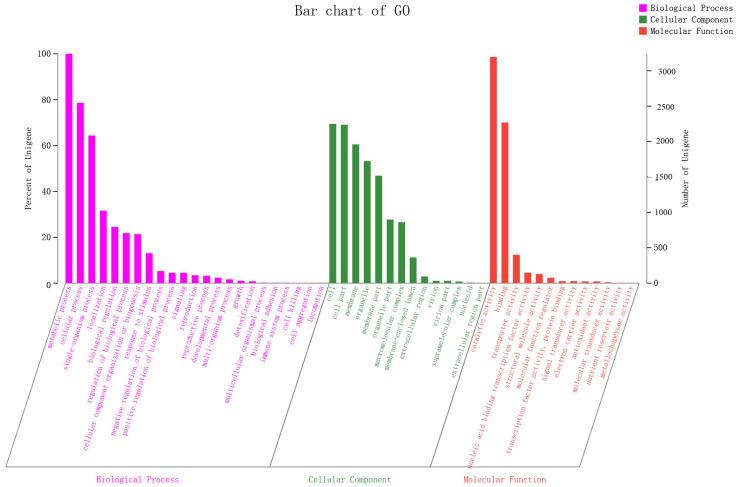
Transcript GO annotation classification statistics graph.

**Figure 9 foods-10-01736-f009:**
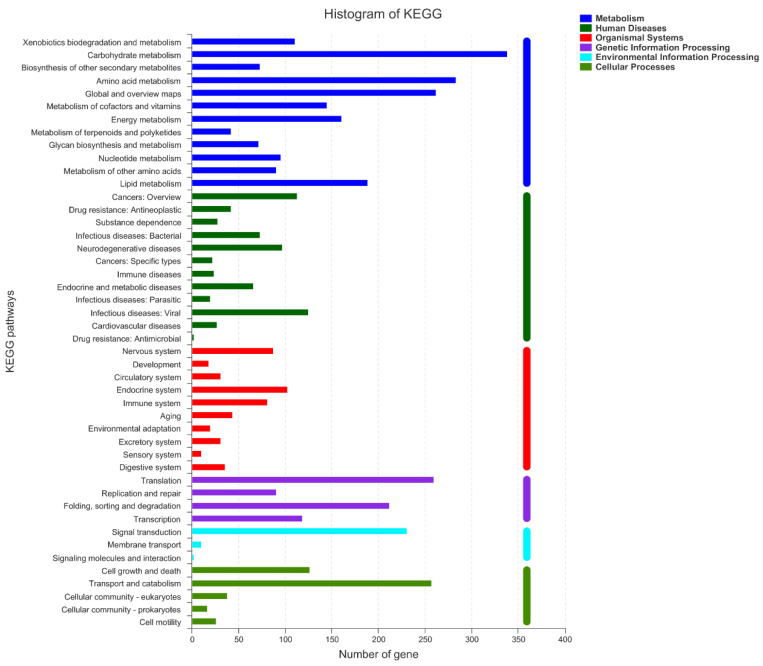
KEGG function classification of Unigenes from *A. japonicas* PJ01.

**Figure 10 foods-10-01736-f010:**
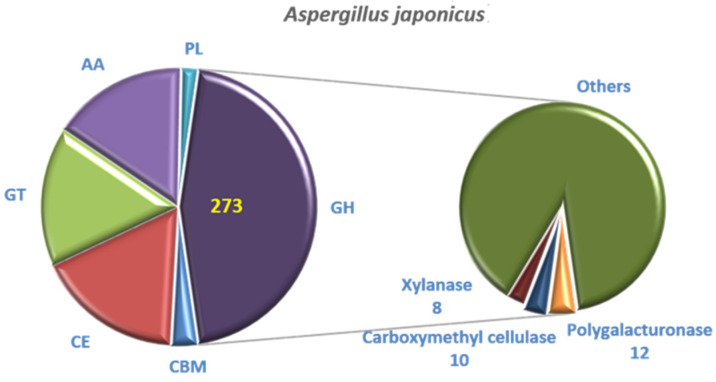
CAZyme annotation result of *A. japonicas* PJ01.

**Table 1 foods-10-01736-t001:** Effect of enzyme solution on micronucleus rate of bone marrow cells in mice (*n* = 5).

Gender	Group	PCE (*n*)	Micronucleus Number (*n*)	Micronucleus Rate (‰)	Nce Number (*n*)	PCE/NCE
♀	C	5000	6	1.20 ± 0.84	883	1.133 ± 0.017
♀	L	5000	7	1.40 ± 1.14	890	1.124 ± 0.016
♀	M	5000	7	1.40 ± 1.14	885	1.130 ± 0.010
♀	H	5000	5	1.20 ± 0.84	892	1.121 ± 0.024
♀	P	5000	150	30.00 ± 3.39 **	954	1.048 ± 0.012 **
♂	C	5000	6	1.20 ± 0.84	883	1.133 ± 0.018
♂	L	5000	7	1.40 ± 1.14	889	1.125 ± 0.015
♂	M	5000	6	1.20 ± 0.84	888	1.127 ± 0.023
♂	H	5000	6	1.20 ± 1.10	879	1.138 ± 0.019
♂	P	5000	146	29.20 ± 1.92 **	944	1.059 ± 0.011 **

Note: Data are expressed as mean ± SD (*n* = 5); “**” means compared with the control group, *p* < 0.01; PCE, polychromatic erythrocytes; NCE, normochromatic erythrocytes; ♂ stands for male and ♀ for female.

**Table 2 foods-10-01736-t002:** Effect of enzyme solution on the incidence of sperm malformation in mice.

Group	Observed	Total Number of Deformities (*n*)	Deformity Rate (%)
	Sperms (*n*)		
Control group	5000	135	2.72 ± 0.29
Low group	5000	133	2.66 ± 0.11
Medium group	5000	129	2.58 ± 0.15
High group	5000	135	2.70 ± 0.34
Positive group	5000	431	8.62 ± 0.83 **

Notes: Data are expressed as mean ± SD (*n* = 5); “**” means compared with the control group, *p* < 0.01.

**Table 3 foods-10-01736-t003:** Effects of enzyme solution on hematology of rats (*n* = 5).

Gender	Test Index		
	(Unit)	Control Group	Enzyme Group
♀	WBC (10^9^/L)	7.01 ± 1.73	6.53 ± 2.78
♀	Neu (%)	5.50 ± 2.91	7.76 ± 2.68
♀	Lym (%)	93.70 ± 3.38	91.34 ± 3.15
♀	Mon	0.46 ± 0.71	0.42 ± 0.44
♀	Eos (%)	0.24 ± 0.23	0.36 ± 0.18
♀	Bas (%)	0.10 ± 0.00	0.12 ± 0.04
♀	RBC (10^12^/L)	7.12 ± 0.41	7.11 ± 0.10
♀	HGB (g/L)	149.80 ± 4.09	151.40 ± 2.41
♀	HCT (%)	41.74 ± 1.25	42.02 ± 0.91
♀	PLT (10^9^/L)	658.20 ± 46.33	690.60 ± 36.56
♂	WBC (10^9^/L)	9.53 ± 2.47	10.41 ± 1.59
♂	Neu (%)	10.26 ± 2.09	9.46 ± 2.36
♂	Lym (%)	86.44 ± 3.73	88.72 ± 3.21
♂	Mon (%)	2.60 ± 1.51	1.54 ± 1.67
♂	Eos (%)	0.60 ± 0.46	0.18 ± 0.08 *
♂	Bas (%)	0.10 ± 0.00	0.10 ± 0.00
♂	RBC (10^12^/L)	7.05 ± 0.24	7.04 ± 0.14
♂	HGB (g/L)	148.40 ± 6.39	153.20 ± 1.10
♂	HCT (%)	41.00 ± 1.85	42.18 ± 0.57
♂	PLT (10^9^/L)	686.60 ± 63.10	676.00 ± 27.86

Notes: Values are given as mean ± SD in each group (*n* = 5); “*” means compared with the control group, *p* < 0.05; ♂ stands for male and ♀ for female. Abbreviations: platelet count (PLT), hemoglobin concentration (HGB), hematocrit (HCT), white blood cell count (WBC) and classification, neutrophil (NEU), lymphocyte (LYM), monocyte (MON), eosinophil (EOS), basophil (BAS) and red blood cell count (RBC).

**Table 4 foods-10-01736-t004:** Effects of test substance on blood biochemical indexes of rats (*n* = 5).

Gender	Test Index		
	(Unit)	Control Group	Enzyme Group
♀	ALB (g/L)	36.40 ± 7.23	35.20 ± 4.60
♀	ALT (U/L)	40.60 ± 9.45	34.60 ± 9.63
♀	AST (U/L)	66.20 ± 15.29	70.00 ± 25.66
♀	BUN (mmol/L)	8.90 ± 1.21	8.63 ± 0.87
♀	CHO (mmol/L)	1.93 ± 0.40	2.14 ± 0.45
♀	Cr (umol/L)	88.20 ± 8.14	83.80 ± 8.53
♀	GLU (mmol/L)	8.29 ± 0.36	7.36 ± 1.85
♀	TG (mmol/L)	0.65 ± 0.19	0.49 ± 0.10
♀	TP (g/L)	67.00 ± 6.71	71.40 ± 7.47
♂	ALB (g/L)	31.00 ± 3.00	31.20 ± 1.64
♂	ALT (U/L)	39.40 ± 10.41	42.20 ± 13.41
♂	AST (U/L)	65.80 ± 16.41	69.60 ± 15.61
♂	BUN (mmol/L)	9.35 ± 1.15	9.66 ± 1.55
♂	CHO (mmol/L)	1.77 ± 0.16	1.98 ± 0.10
♂	Cr (umol/L)	70.80 ± 11.45	70.40 ± 15.31
♂	GLU (mmol/L)	7.99 ± 1.50	8.12 ± 0.69
♂	TG (mmol/L)	0.69 ± 0.18	0.84 ± 0.11
♂	TP (g/L)	56.40 ± 10.55	69.80 ± 15.06

Notes: Values are given as mean ± SD in each group (*n* = 5). ♂ stands for male and ♀ for female. Abbreviations: total protein (TP), albumin (ALB), blood urea nitrogen (BUN), blood glucose (GLU), alanine aminotransferase (ALT), aspartate aminotransferase (AST), cholesterol (CHO), triglyceride (TG), creatinine (Cr).

**Table 5 foods-10-01736-t005:** The genome features of *A. japonicas* PJ01.

Attributes	Characteristic
Genome size (bp)	36204647
G+C content (%)	51.37%
GC content in gene region (%)	54.84%
GC content in intergenetic region (%)	46.28%
Protein-coding genes (CDS)	10070
Gene total len (bp)	21519089
Gene/genome (%)	59.44%
Intergenetic region len (bp)	14685558
Intergenetic len/genome (%)	40.56%
tRNA genes	351
5S rRNA	43
5.8S rRNA	26
18S rRNA	0
28S rRNA	24
Genes assigned to NR	10070
Genes assigned to Swiss-Prot	6891
Genes assigned to Pfam	7392
Genes assigned to COG	8489
Genes assigned to GO	5955
Genes assigned to KEGG	3303
Genes assigned to CAZy	601

## Data Availability

The data presented in this study are available in this article. The whole genome sequencing data in this study can be found in the GenBank database of the NCBI website with the accession number of PRJNA715308. The link: http://www.ncbi.nlm.nih.gov/bioproject/715308, accessed on 17 May 2021.

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
