# Peer review of "Safety Evaluation and Whole Genome Sequencing of Aspergillus japonicas PJ01 Reveal Its Potential to Degrade Citrus Segments in Juice Processing"

_foods, 2021, doi:10.3390/foods10081736_

Round 1
Reviewer 1 Report
The authors sequenced the genome of PJ01. However, I find that the manuscript is preliminary and a bit disoriented. For instance, the first two paragraphs of the introduction are general statements, not generating questions to build a manuscript. I am not convinced regarding the safety statement of PJ01. The method of crude enzyme preparation is not clear. Although it was filtered, the method does not explain if the resulting enzyme would be sterile. This seriously affects the use of enzyme as food. I can see several grammatical mistakes which could be improved with professional English editing, but what I am concerned is that I can see many scientific names are not italicized, indicating the authors were not serious while preparing the manuscript draft.
How was statistical analysis performed? Which tools were used? This needs to be explained in the method section. Table 1 and table 3: A higher SD compared to mean indicate no normal distribution of the data indicating that the comparisons would not be meaningful.
I have few other comments regarding this manuscript.
2.2: Production of crude enzyme: This is the enzyme used for the experiments in this manuscript. This section needs details rather than citations.
2.5: This manuscript is regarding genome assembly. The detailed method of assembly, including the number of reads, assembly tools used (including version) needs to be explained. I don’t understand the sentence “……..the filtered reads were assembled”
Effect of crude enzyme solution on the degradation of Citrus segments: I’m sorry if I didn’t understand. The image is showing fruit. Petal is a part of a flower, not fruit.
Line 316: I am not convinced regarding the safety and non toxic as the data were not normally distributed.
Author Response
Dear reviewer, we have explained the series of questions you raised below and revised them in the manuscript.
1.The authors sequenced the genome of PJ01. However, I find that the manuscript is preliminary and a bit disoriented. For instance, the first two paragraphs of the introduction are general statements, not generating questions to build a manuscript. I am not convinced regarding the safety statement of PJ01. The method of crude enzyme preparation is not clear. Although it was filtered, the method does not explain if the resulting enzyme would be sterile. This seriously affects the use of enzyme as food. I can see several grammatical mistakes which could be improved with professional English editing, but what I am concerned is that I can see many scientific names are not italicized, indicating the authors were not serious while preparing the manuscript draft.
Response: Thank you for your comments for our manuscript. The first two paragraphs are mainly for people to understand the prospect and application of the enzyme solution, the significance and innovation of our research, and we have made supplement and modification in the manuscript (lines 63-73). We added the content of the method of crude enzyme preparation in 2.2 (lines 89-94). The whole process is operated in the sterile room. The spore suspension of Aspergillus japonicus PJ01 is transferred to the medium after high-pressure sterilization for fermentation to produce enzyme without adding other bacteria. We will try our best for this manuscript. We will try to correct the deficiencies.
2.How was statistical analysis performed? Which tools were used? This needs to be explained in the method section. Table 1 and table 3: A higher SD compared to mean indicate no normal distribution of the data indicating that the comparisons would not be meaningful.
Response: Thank you for your comments for our manuscript. Excel and SPSS 26 software were used for statistical analysis. The measurement data were tested for normality and homogeneity of variance. If the normality (P > 0.05) was satisfied, one-way ANOVA was used for statistical analysis, and LSD + Dunnet or tamhane's T2 was selected for comparative analysis according to the homogeneity of variance. If the normality is not satisfied (P ≤ 0.05), Mann-Whitney U test is used for pairwise comparison analysis, where p ≤ 0.05 means statistically significant and P ≤ 0.01 means highly statistically significant. In Table 1, because the number of micronucleus cells is relatively small, some even zero, so the value of SD is large. The enzyme solution had little effect on micronucleus cells. We explained Table 3 in particular in question 6, please check it.
- Production of crude enzyme: This is the enzyme used for the experiments in this manuscript. This section needs details rather than citations.
Response: Thank you for your comments for our manuscript. We added the content of the method of crude enzyme preparation in 2.2 (lines 89-94). Aspergillus japonicus PJ01 was cultured at 35 °C for three days, and then washed with normal saline containing 0.1% Tween 80 to prepare spore suspension (1×106/ml). Using 75 mL modified Czapek’s medium (g/L) (NaNO3 5, K2HPO4 1, KCL 0.5, MgSO4·7H2O 0.5, and FeSO4·7H2O 0.01), 1% orange peel powder and 0.35 g PEG4000, 1 ml spore suspension was inoculated in it after high pressure sterilization. Then it was cultured at 35 °C for 72 hours. The supernatant after filtration and centrifugation was used as crude enzyme solution.
- This manuscript is regarding genome assembly. The detailed method of assembly, including the number of reads, assembly tools used (including version) needs to be explained. I don’t understand the sentence “…….. the filtered reads were assembled”
Response: Thank you for your comments for our manuscript. Referring to other people's methods, we revised and improved the manuscript in 2.5 (lines 176-198).
5.Effect of crude enzyme solution on the degradation of Citrus segments: I’m sorry if I didn’t understand. The image is showing fruit. Petal is a part of a flower, not fruit.
Response: Thank you for your comments for our manuscript. We used orange slices here. Maybe the previous expression is ambiguous and we have modified it. Our main purpose is to study the degradation of segments outside orange slices by the crude enzyme solution. The first two paragraphs of the introduction described it. It is mainly expected to be used in the production of citrus juice and canned orange.
6.Line 316: I am not convinced regarding the safety and non-toxic as the data were not normally distributed.
Response: Thank you for your comments for our manuscript. The study was reviewed and approved by the Institutional Animal Care and Use Com-mittee (IACUC) of Hunan Sileike Jingda Co. with certificate no. IACUC-SJA18072. We conducted the safety evaluation in a qualified biological company, and they issued a professional safety evaluation report to prove that the crude enzyme solution of Aspergillus japonicus PJ01 is safe. Among them, MON %and EOS % with larger SD are specially described. The value of Hematology itself is a percentage, not a specific value, so there may be a situation where the standard deviation is greater than the average value. If the normality is not satisfied (P ≤ 0.05), Mann-Whitney U test is used for pairwise comparison analysis, where p ≤ 0.05 means statistical significance and P ≤ 0.01 means highly statistical significance. In the following, we test and calculate MON and EOS. Because the data of control group does not conform to the normality, we use pairwise comparison. The EOS of enzyme group was significantly different from that of control group in male (P < 0.05), but it was within the normal range and had no physical significance.
|
Normality test |
||||||||
|
|
Group |
Kolmogorov sminov (V)a |
Shapiro Wilke |
|||||
|
|
Statistics |
freedom |
Significance |
Statistics |
freedom |
Significance |
||
MON% |
Control group |
Female |
.334 |
5 |
.072 |
.740 |
5 |
.024 |
|
Male |
.205 |
5 |
.200* |
.936 |
5 |
.634 |
|||
Enzyme group |
Female |
.290 |
5 |
.197 |
.795 |
5 |
.073 |
||
Male |
.380 |
5 |
.017 |
.689 |
5 |
.007 |
|||
EOS% |
Control group |
Female |
.197 |
5 |
.200* |
.943 |
5 |
.685 |
|
Male |
.386 |
5 |
.014 |
.723 |
5 |
.016 |
|||
Enzyme group |
Female |
.229 |
5 |
.200* |
.867 |
5 |
.254 |
||
Male |
.231 |
5 |
.200* |
.881 |
5 |
.314 |
|||
|
*. This is the lower limit of true significance. |
||||||||
|
a. Riley's significance correction. |
||||||||
Test Statisticsa |
||
|
Mon% |
Eos% |
Mann-Whitney U |
11.500 |
8.000 |
Wilcoxon W |
26.500 |
23.000 |
Z |
-.213 |
-.958 |
Asymp. Sig. (2-tailed) |
.831 |
.338 |
Exact Sig. [2*(1-tailed Sig.)] |
.841b |
.421b |
a. Grouping: Female in control and enzyme group |
||
b. Not corrected for ties. |
Test Statisticsa |
||
|
Mon% |
Eos% |
Mann-Whitney U |
9.000 |
1.000 |
Wilcoxon W |
24.000 |
16.000 |
Z |
-.731 |
-2.455 |
Asymp. Sig. (2-tailed) |
.465 |
.014 |
Exact Sig. [2*(1-tailed Sig.)] |
.548b |
.016b |
a. Grouping Variable: Male in control and enzyme group |
||
b. Not corrected for ties. |
Finally, thank you very much for your valuable comments on our manuscript, which has given us great help. We uploaded the revised manuscript with tracked changes to highlight the revisions. However, due to the slight difference the number of lines between manuscript marked with tracking modification and the original version, the number of lines mentioned in above is in the case of opening the mark. If there are other shortcomings, please continue to urge us to revise.

Reviewer 2 Report
The manuscript # foods-1266305, by Qian et al, entitled: Safety evaluation and whole genome sequencing of Aspergillus japonicas PJ01 reveal its potential to degrade citrus segments in juice processing of particular interest. It is an innovative study that aims to investigate the enzyme production ability of Aspergillus strain used in juice processing. The data are interesting but in the present form multiple aspects remains unclear which need to be resolved:
For the acute toxicity test, the authors are not indicating the rational for choosing the dose of 40ml/kg. It is not clear how they concluded that this was the maximum tolerated dose (MTD). The authors used the body weight as the only outcome to determine the acute toxicity, it would have also been interesting to evaluate other signs of toxicity.
Although a negative control was used, the authors are missing a second control that could be an additional strain of Aspergillus to see if the effect is strain specific.
For the micronucleus test, A need negative (non-exposed) and positive (exposed to a well-characterized toxin) controls are needed, the authors are missing a positive control for their study. The test needs to be completed using multiple doses that should be administered withing the 24-48 hours of the test. The authurs are missing all these details in the study.
The COG function classifications and KEGG_pathway analysis are very interesting and are showing the potential of the strain, however, it is not clear what are the strain specific pathways, and enzyme, it would have been more interesting if the authors conducted a genomic comparative analysis, or at least conducting a multiple sequence analysis of genes coding for the enzyme of interest.
Author Response
Dear reviewer, we have explained the series of questions you raised below and revised them in the manuscript.
- For the acute toxicity test, the authors are not indicating the rational for choosing the dose of 40ml/kg. It is not clear how they concluded that this was the maximum tolerated dose (MTD). The authors used the body weight as the only outcome to determine the acute toxicity, it would have also been interesting to evaluate other signs of toxicity.
Response: Thank you for your comments for our manuscript. Horne's method is the most common traditional method for acute oral toxicity test. When the solvents are water, oil or alcohol, the intragastric volume of each test group is the same. The intragastric volume of rats is 40 ml / kg, and that of mice is 20 ml / kg. When the solvent is distilled water, the maximum intragastric volume of rats is 80 ml / kg, and that of mice is 40 ml / kg. According to Horne's method, we set the maximum intragastric volume of mice was 40 ml / kg. A number of daily observations were made on mice in acute oral toxicity test. The daily observations of the mice included: appearance, behavioral activities, secretions, excretions, diet, and death (time of death, pre-death reaction). The body weight of each mouse was recorded on day 0, 7, 10 and 14 of administration (lines 126-128). After 14 days of continuous observation, no abnormal reaction and death were found. At the end of the experiment, gross anatomy was conducted on mice, and no obvious abnormal conditions were found on the surface and section of main organs (lines 225-227).
2.Although a negative control was used, the authors are missing a second control that could be an additional strain of Aspergillus to see if the effect is strain specific.
Response: Thank you very much for your valuable comments on our manuscript. The fungal strain used in this study was Aspergillus japonicas PJ01 (CCTCC NO: M2013323), isolated from the rotten branches from in Yuelu Mountain soil. It was kept in China Center for Type Culture Collection (CCTCC). In this manuscript, we investigated the functional effect of the enzyme produced by it on the degradation of Citrus capsule, hoping to apply it to juice processing and canned production. Safety evaluation is to verify the safety of its application in food processing. Gene sequencing is to find the gene fragment of enzyme production, which is conducive to the development of genetic engineering and enzyme engineering, and promote the industrialization of enzyme production. In the future experiments, we will further investigate the differences between its ability and other strains.
3.For the micronucleus test, A need negative (non-exposed) and positive (exposed to a well-characterized toxin) controls are needed, the authors are missing a positive control for their study. The test needs to be completed using multiple doses that should be administered withing the 24-48 hours of the test. The authors are missing all these details in the study.
Response: Thank you for your comments for our manuscript. We will try our best for this manuscript. Sterile distilled water was administered to the negative control group and the positive control group was treated with 40 mg/kg cyclophosphamide. Using the method of 30 h administration of test substance, the interval between the two treatments was 24 hours, and the mice were killed six hours after the second treatment for bone marrow cell staining (lines 139-141).
4.The COG function classifications and KEGG_pathway analysis are very interesting and are showing the potential of the strain, however, it is not clear what are the strain specific pathways, and enzyme, it would have been more interesting if the authors conducted a genomic comparative analysis, or at least conducting a multiple sequence analysis of genes coding for the enzyme of interest.
Response: Thank you so much for your comment and suggestions. We have changed this part of the content, but consulted a lot of literature, there is little information about the specific gene sequence coding of the enzyme, and it is a little difficult to compare with its sequence. In the next study, we will compare the enzyme production ability and genome sequencing of the three strains isolated before.
Finally, thank you very much for your valuable comments on our manuscript, which has given us great help. We uploaded the revised manuscript with tracked changes to highlight the revisions. However, due to the slight difference the number of lines between manuscript marked with tracking modification and the original version, the number of lines mentioned above is in the case of opening the mark. If there are other shortcomings, please continue to urge us to revise.

Reviewer 3 Report
The authors reported the safety evaluation of Aspergillus japonicas Pj01, for food processing applications.
The manuscript is well written and the results are clearly presented. The discussion and the conclusions are well related to the presented results.
Minor Revision:
Methods Lines 153-161: Please better clarify how reads were filtered (indicate software and parameters); indicate also software parameters or indicate "default parameters were used". Please, also indicate how error correction was performed. Finally, provide appropriate references for software and database used (lines 158-161)
Results, Lines 257-260. "has more relationship". I would not use such an expression. Better, a more neutral form as: "most abundant COGs are (in decreasing order): carbohydrate transport, post translational, ..."
Results, line269. I do not believe "demonstrated" is the correct verb for this purpose, better: "returned". Also, "enriched", better "annotated".
Results, line 286. same of the above comment, please substitute "demonstrated" with "returned".
Results, line 289. In this analysis, it is not correct to speak in terms of abundance. It is better to speak in terms of genes assigned to a given pathway. As example: three hundred thirty-nine genes were assigned to the carbohydrate metabolism, followed by the amino acid metabolism with 284 genes,.... This aspect further appears in discussion (lines 345-349) In KEGG annotation, many genes are related... It is better to use a different form: KEGG-based analysis highlighted a large number of genes involved in carbohydrate metabolism, among which 52 are related to starch....
Finally: there are many typos: spaces between words and commas, please check them carefully. Please check carefully references, some of them did not seem related to the text
Author Response
Dear reviewer, we have explained the series of questions you raised below and revised them in the manuscript.
1.Methods Lines 153-161: Please better clarify how reads were filtered (indicate software and parameters); indicate also software parameters or indicate "default parameters were used". Please, also indicate how error correction was performed. Finally, provide appropriate references for software and database used (lines 158-161)
Answer: Thank you for your comments for our manuscript. We revised in 2.5 Whole genome sequencing, assembly and annotation. The analysis software is described in more detail. Please refer to the manuscript for details (lines 176-198).
2.Results, Lines 257-260. "has more relationship". I would not use such an expression. Better, a more neutral form as: "most abundant COGs are (in decreasing order): carbohydrate transport, post translational, ..."
Answer: Thank you for your comments for our manuscript. We revised it in lines 297-300.
3.Results, line269. I do not believe "demonstrated" is the correct verb for this purpose, better: "returned". Also, "enriched", better "annotated". line 286. same of the above comment, please substitute "demonstrated" with "returned".
Answer: Thank you for your comments for our manuscript. We have revised it in the manuscript (line 309 and 327).
4.Results, line 289. In this analysis, it is not correct to speak in terms of abundance. It is better to speak in terms of genes assigned to a given pathway. As example: three hundred thirty-nine genes were assigned to the carbohydrate metabolism, followed by the amino acid metabolism with 284 genes,.... This aspect further appears in discussion (lines 345-349) In KEGG annotation, many genes are related...
Answer: Thank you for your comments for our manuscript. We have revised it in the manuscript (line 331 and lines 391-395).
5.Finally: there are many typos: spaces between words and commas, please check them carefully. Please check carefully references, some of them did not seem related to the text
Answer: Thank you for your comments for our manuscript. We have revised it earnestly. Some adjustments have been made to the references.
Finally, thank you very much for your valuable comments on our manuscript, which has given us great help. We uploaded the revised manuscript with tracked changes to highlight the revisions. However, due to the slight difference the number of lines between manuscript marked with tracking modification and the original version, the number of lines mentioned above is in the case of opening the mark. If there are other shortcomings, please continue to urge us to revise.

Round 2
Reviewer 1 Report
The authors did a great job revising the paper. Regarding the scientific merit, I don't have problems recommending acceptance.
Another severe issue I am having problems is:
The authors mention that data is private and not publicly available upon acceptance (Line 427).
I want to make sure that the following are uploaded to NCBI and made available to the public.
- Raw or corrected Illumina reads
- Raw or corrected pacbio reads
- Final assembly
Another very minor issue I found again-
- Scientific names are not italicized at some points.
- Once you write the complete genus and species name, the genus name can be abbreviated on the second use. Please correct it during revision/proofreading process.
Author Response
Dear reviewer,
1.We upload all the data of genome sequencing to the database, and after the manuscript is received, we will release the data and make it public.
2.Thank you for your comments and your efforts in our manuscript. We made changes in the manuscript. Aspergillus japonicus PJ01 was abbreviated to A. japonicus PJ01 on its second use.
This manuscript is a resubmission of an earlier submission. The following is a list of the peer review reports and author responses from that submission.